# Evaluation of a semiquantitative SNAP test for measurement of bile acids in dogs

Rachel L. Seibert[1], Karen M. Tobias[1], Ann Reed[2] and Karl R. Snyder[3]

[1] Department of Small Animal Clinical Sciences, College of Veterinary Medicine, University of Tennessee, Knoxville, TN, USA
[2] Office of Information Technology, University of Tennessee, Knoxville, TN, USA
[3] Department of Pathology, College of Veterinary Medicine, University of Tennessee, Knoxville, TN, USA

## ABSTRACT

**Background.** Serum bile acids (SBA) are used as a routine screening tool of liver function in dogs. Serum samples are usually shipped to a referral laboratory for quantitative analysis with an enzymatic chemistry analyzer. The canine SNAP Bile Acids Test (SNAP-BAT) provides an immediate, semi-quantitative measurement of bile acid concentrations in-house. With the SNAP-BAT, bile acids concentrations of 5–30 µmol/L are quantified, and results outside of that range are classified as <5 or >30 µmol/L. Agreement of the SNAP-BAT with the enzymatic method has not been extensively investigated.

**Objectives.** The purposes of this prospective clinical study were to assess the precision of the SNAP-BAT and determine agreement of SNAP-BAT with results from an in-house chemistry analyzer.

**Methods.** After verifying intra-assay precision of the SNAP-BAT, a prospective analysis was performed using blood samples collected from 56 dogs suspected to have liver disease. Each sample was analyzed with an enzymatic, in-house chemistry analyzer and the SNAP-BAT. Agreement between the two methods was statistically assessed using the $\kappa$ index of agreement.

**Results.** Intra-assay variability was minimal. The $\kappa$ index for agreement between the SNAP-BAT and routine chemistry analyzer was between 0.752 and 0.819, indicating substantial to near perfect agreement.

**Conclusions.** The SNAP-BAT is a highly accurate, semi-quantitative test that yields immediate results, and has very little intra-assay variability, particularly for results >30 µmol/L.

Corresponding author
Rachel L. Seibert, rseiber2@utk.edu

## INTRODUCTION

Bile acids are synthesized exclusively within hepatocytes from cholesterol, then excreted into the biliary tract and stored in the gallbladder. After a meal, bile acids are released into the duodenum, where they facilitate lipid digestion and absorption. Over 95% of bile acids are reabsorbed in the distal ileum and jejunum and transported, via the portal circulation, to the liver. Within the liver, bile acids are removed from portal blood by hepatocytes and recycled back into the biliary system (termed enterohepatic circulation).

An increase in SBA can therefore be expected with hepatic, biliary, or portal disorders that limit hepatic portal blood flow or hepatocellular uptake (*Jensen, 1991*). In dogs, pre- and postprandial SBA have repeatedly been found to be reliable markers for the diagnosis of hepatic and biliary dysfunction and disorders of the portal vasculature, such as portosystemic shunts (PSS) and congenital portal vein hypoplasia with secondary microvascular dysplasia (PVH-MVD). In previous studies of dogs with PSS, sensitivity of increased SBA concentration for diagnosis of PSS ranged from 64% to 100% (*Winkler et al., 2003*; *Center et al., 1985a*; *Meyer, 1986*; *Gerritzen-Bruning, van den Ingh & Rothuizen, 1985*). The specificity of SBA for the diagnosis of liver disease exceeds 90% at SBA concentrations greater than or equal to 30 µmol/L and reaches 100% at concentrations greater than or equal to 50 µmol/L (*Center et al., 1985b*).

The clinical standard for measurement of bile acids is considered to be an enzyme-based assay, which provides fully quantitative results. However, most veterinary practices do not have this methodology available in-house. Samples must be sent off to an outside laboratory, resulting in delays in diagnosis and treatment. Recently, a rapid, in-house bile acid test—the IDEXX Canine Snap Bile Acids Test [SNAP-BAT] (IDEXX Laboratories, Inc., Sacramento, CA)—has become commercially available. The SNAP-BAT is a semi-quantitative, competitive immunoassay that fully quantifies values of 5–30 µmoles/L and yields semiquantitative results for bile acids <5 µmoles/L and >30 µmoles/L. This test measures the concentration of bile acids using an antibody present within the reagent (or conjugate), which is mixed with the serum sample. The conjugate contains an antibody to bile acids with an enzyme linked to it (termed "labeled") , which allow for detection of bile acids by producing an observable color change when the labeled antibody binds to bile acids. Because this is an in-house test, the SBAT is becoming more widely used by private practitioners as a screening test for dogs with clinical signs consistent with portal vascular anomalies and other liver diseases. In a method comparison study by the manufacturer, overall agreement of the SNAP-BAT with the reference assay (Hitachi Chemistry Analyzer) was 96.2% for bile acid concentrations <12 µmol/L and 95.2% for bile acid concentrations >25 µmol/L (*Holbrook, Roth-Johnson & Sheldon, 2005*). However, statistical assessment of agreement between the two tests and calculation of test sensitivity and specificity for diagnosis of liver dysfunction were not performed in that study. Additionally, evaluation of the new SNAP-BAT was performed by the manufacturer; independent analysis by a third party has not been reported.

The purposes of this prospective clinical study were to assess the precision, or intra-assay variability, of the SNAP-BAT and determine agreement of SNAP-BAT with results from a standard in-house enzymatic chemistry analyzer (Cobas 501c; Roche Diagnostics, Indianapolis, IN). We hypothesized intra-assay variability of the SNAP-BAT would be minimal and there would be no statistically significant difference in results of SBAT and the in-house, enzymatic chemistry analyzer for quantitative and semiquantitative data.

## MATERIALS AND METHODS

### Samples

All procedures were approved by the University of Tennessee Institutional Animal Care and Use Committee (Number: 2053-1011) before commencement of the study, and informed consent was obtained from owners of all enrolled dogs. Analysis was performed on blood drawn for enzymatic bile acid analysis from patients at the University of Tennessee or serum submitted by local practices to the University of Tennessee Clinical Pathology Laboratory. Samples obtained in-house were classified as preprandial or postprandial. Preprandial samples were obtained after a fast of 12 or more hours, and postprandial samples were obtained 2 h after feeding. Samples were collected by cephalic, saphenous, or jugular venipuncture from in-house patients. Blood was placed in a red top tube containing no additives, allowed to clot for a minimum of 20 min, and then centrifuged for 10 min at 2250 g (Clay Adams Brand Compact II Centrifuge; Becton Dickinson Primary Care Diagnostics, Sparks, MD). The serum was then separated and stored at −20 °C until the time of analysis. Because specific fasting and feeding information was not available, samples submitted from local practices were considered unclassified. Serum samples submitted by local practices were also similarly centrifuged, separated from any remaining red cells, and stored at −20 °C. Maximum storage time was 3 months.

### Sample analysis

At the time of biochemical analysis, frozen samples were thawed at room temperature and gently agitated until all crystals were dissolved. All SNAP-BAT were performed by the same evaluator (RS) using the methodology described by the manufacturer (IDEXX Laboratories, Inc., Sacramento, CA) with interpretation of colorimetric changes provided by the SNAP Reader (IDEXX Laboratories, Inc., Sacramento, CA). The standard enzymatic chemical analysis was performed by the same clinical pathology technician (KS).

#### *Intra-assay variability*

Five frozen serum samples were randomly selected, thawed as described above, and combined into a pooled sample. Each pooled sample was evaluated with the standard chemistry analyzer (Cobas 501c; Roche Diagnostics, Indianapolis, IN) using an enzyme-based reagent (Diazyme Laboratories, Poway, CA). Each pooled sample was then divided into 10 allotments, and each allotment was evaluated with a SNAP-BAT. This process was repeated 4 more times to generate a total of 5 pooled samples, resulting in a total of 50 SNAP-BAT and 5 standard, enzymatic chemistry analyzer tests. All sample analyses were performed on the same day to reduce testing variables.

#### *Quantitative analysis*

After confirmation of minimal intra-assay variability, quantitative analysis was performed on samples different from those used for the intra-assay variability testing. Frozen serum samples were thawed as described above. One aliquot (0.5 ml) from each sample was evaluated using the standard, enzymatic chemistry analyzer. The remaining serum from each sample was evaluated on the same day using the SNAP-BAT.

## Statistical analysis

For general agreement, bile acid concentrations measured by SNAP-BAT assay were compared with the results of enzymatic analysis using the $\kappa$ index (*Landis & Koch, 1977*; *Cohen, 1960*). Reference range for normal bile acids was set at 0–20 μmol/L, which is the reference range currently used by our laboratory for results of enzymatic analysis. This reference range was used regardless of whether the sample was pre- or postprandial (*Center, 1993*). Calculations of the $\kappa$ index sensitivity, specificity, and negative and positive predictive values of the SNAP-BAT were therefore calculated based on a cutoff value of 20 μmol/L. Data analysis was performed using two groupings: Group 1, which included results from dogs with postprandial samples and those with a single, unclassified serum sample; and Group 2, which included results from preprandial samples and dogs with a single, unclassified serum sample. Each analysis therefore included one observation per dog to avoid pseudoreplication of results.

Each analysis was further divided into two sets of data: bile acid concentrations $\leq 20$ μmol/L or $>20$ μmol/L, and bile acid concentrations $<5$ μmol/L, 5–30 μmol/L, and $>30$μmol/L. Because variances were not equal, the nonparametric Mann–Whitney test was used to test differences in enzymatic and SNAP-BAT results using two SNAP categories (SNAP-BAT $\leq 20$ μmol/L; SNAP-BAT $> 20$ μmol/L). The Kruskal–Wallis test was used to compare three categories of results (SNAP-BAT $< 5$ μmol/L; 5–30 μmol/L, and $>30$ μmol/L) with those of the enzymatic results for the same sample. A P value $<0.05$ was used to indicate statistical significance.

## RESULTS

Samples from 25 dogs were used for evaluation of intra-assay variability. Intra-assay variability was minimal overall, with no variability seen in samples with values $>30$ μmol/L (Table 1). Because of lack of variability and small sample size, statistical analysis of these results was unable to be performed.

Samples from 56 dogs were included in the quantitative analysis. Paired (pre- and post-prandial) bile acid samples were available from 17 dogs, and a single, unclassified sample was available from each of 39 dogs. The cross tabulation data for results of enzymatic chemistry analysis and SNAP-BAT for Group 1 and Group 2 are reported in Tables 2 and 3, respectively. For Group 1 results, sensitivity of the SNAP-BAT was 100% and specificity was 78.6%. The positive predictive was 82.4%, while the negative predictive value was 100%. For Group 2 results, sensitivity of SNAP-BAT was 100% and specificity was 83.9%. The positive predictive value was 83.3% and negative predictive value was 100%.

For Group 1 samples, the $\kappa$ index for agreement between the SNAP-BAT and routine chemistry analyzer was 0.786. For Group 2 samples, the $\kappa$ index for agreement between the SNAP-BAT and routine chemistry analyzer was 0.819 (Table 4). When enzymatic chemistry values were ranked based on two SNAP-BAT categories ($\leq 20$ and $>20$ μmol/L) using the Mann–Whitney test, the values differed significantly from each other between the two categories for both Group 1 and Group 2 (Mann–Whitney U $= 31.5$, $p < 0.001$ for Group 1; Mann–Whitney U $= 25.5$, $p < 0.001$ for Group 2). Similarly, when

**Table 1  Intra-assay variability for SNAP-BAT in µmoles/L.**

|  | Pooled samples | | | | |
|---|---|---|---|---|---|
|  | 1 | 2 | 3 | 4 | 5 |
| Chemistry analyzer value | 87 | 97 | 8 | 10 | 67 |
| SNAP 1 | >30 | >30 | 8 | 11 | >30 |
| SNAP 2 | >30 | >30 | 10 | 17 | >30 |
| SNAP 3 | >30 | >30 | 7 | 13 | >30 |
| SNAP 4 | >30 | >30 | 10 | 8 | >30 |
| SNAP 5 | >30 | >30 | 12 | <5 | >30 |
| SNAP 6 | >30 | >30 | <5 | 12 | >30 |
| SNAP 7 | >30 | >30 | <5 | 10 | >30 |
| SNAP 8 | >30 | >30 | <5 | 6 | >30 |
| SNAP 9 | >30 | >30 | 9 | 10 | >30 |
| SNAP 10 | >30 | >30 | 7 | 11 | >30 |

**Table 2  Agreement of Group 1 enzymatic chemistry analysis and SNAP-BAT for results categorized as >20 µmoles/L or ≤20 µmol/L.**

|  |  |  | SNAP values[*] | | |
|---|---|---|---|---|---|
|  |  |  | >20 | ≤20 | Total |
| Chemistry values[*] | >20 | Count | 28 | 0 | 28 |
|  |  | % within category | 100.0% | 0% | 100.0% |
|  | ≤20 | Count | 6 | 22 | 28 |
|  |  | % within category | 21.4% | 78.6% | 100.0% |
| Total |  | Count | 34 | 22 | 56 |
|  |  | % within category | 60.7% | 39.3% | 100.0% |

Notes.

[*] Results are in µmoles/L.

**Table 3  Agreement of Group 2 enzymatic chemistry analysis and SNAP-BAT for results categorized as >20 µmoles/L or ≤20 µmol/L.**

|  |  |  | SNAP values[*] | | |
|---|---|---|---|---|---|
|  |  |  | >20 | ≤20 | Total |
| Chemistry values[*] | >20 | Count | 25 | 0 | 25 |
|  |  | % within category | 100% | 0% | 100.0% |
|  | ≤20 | Count | 5 | 26 | 31 |
|  |  | % within category | 16.1% | 83.9% | 100.0% |
| Total |  | Count | 30 | 26 | 56 |
|  |  | % within category | 53.6% | 46.4% | 100.0% |

Notes.

[*] Results are in µmoles/L.

**Table 4 Comparisons of level of agreement and standard error between the enzymatic chemistry analyzer and SNAP-BAT for Groups 1 and 2.**

|  | *k index | SE |
|---|---|---|
| Group 1 | 0.786 | 0.081 |
| Group 2 | 0.819 | 0.076 |

**Notes.**

* Interpretation of k index: 0, no agreement; 0–0.2, slight agreement; 0.21–0.40, fair agreement; 0.41–0.60, moderate agreement; 0.61–0.80, substantial agreement; 0.8, almost perfect agreement (*Landis & Koch, 1977*); SE, Standard error.

enzymatic chemistry values were ranked based on three SNAP-BAT categories ($<5$; 5–30; $>30$ µmol/L) using the Kruskal–Wallis test , the values differed significantly from each other between the three categories for both Group 1 and Group 2 (Chi-square $= 41.064$; degrees of freedom $= 2$; $p < 0.001$ for Group 1; Chi-square $= 40.38$; degrees of freedom $= 2$, $p < 0.001$ for Group 2).

## DISCUSSION

The main purpose of this study was to evaluate the accuracy of the SNAP-BAT for classification of bile acid concentrations as normal or increased. The accuracy of a measurement system is defined as the degree of closeness to the true value. Similar to the method comparison study by the manufacturer, the enzymatic chemistry analyzer used at our institution was considered the gold standard, or reference method, for determination of the true value (*Holbrook, Roth-Johnson & Sheldon, 2005*). To minimize variables, serum samples were frozen and stored after collection so that all testing could be performed on the same day, immediately after thawing. Bile acids can be stored frozen at $-20\,°C$ for at least 3 months with little variability (*Jensen, 1991*; *Olsson, 1988*).

To avoid pseudoreplication, statistical analysis was compared twice, combining results from dogs with a single sample and the preprandial or postprandial results of dogs with paired samples. Feeding status of dogs with single samples was not critical to data analysis, since each patient served as its own control. Results of statistical analysis from both groups were similar. A single reference range of 0–20 µmols/L was used for both groups to account for variations in interictal gallbladder activity: results of 20% of preprandial samples are reportedly greater than their paired postprandial sample because of spontaneous gall bladder contraction (*Pusterla et al., 2002*).

There are limited methods available for statistically comparing semiquantitative to fully quantitative data. Because one method reports discrete data, while the other reports data on a continuous scale, a medically applicable cutoff value must be used in order to determine agreement between methods (20 µmols/L in this case). The $\kappa$ index, used as the statistical method of assessing agreement in the current study, varied between 0.786 and 0.819, confirming a substantial to near perfect level of agreement between the enzymatic chemistry analyzer and the SNAP-BAT, based on statistical standards (*Landis & Koch, 1977*). The overall sensitivity of the SNAP-BAT is 100%; specificity was slightly less, ranging from 78.6% to 83.9%. Because of its high sensitivity, the SNAP-BAT will be useful for detecting liver dysfunction in affected dogs. However, it will yield more false

positive results than enzymatic analysis; thus, healthy dogs could be falsely diagnosed with abnormal liver function. In dogs with a SNAP-BAT result >20 µmol/L, repeating the test with an enzymatic chemistry analyzer would be therefor be necessary to verify the presence of liver dysfunction.

Agreement between the SNAP-BAT and chemistry analyzer was further evaluated using two categories (≤20 µmol/L and >20), and three categories (<5; 5–30; >30). The latter categorization was chosen to reflect the semi-quantitative values given by the SNAP-BAT. The results indicated that values differed significantly between categories, whether there were two or three categories. This further supports the strong agreement between enzymatic and SNAP-BAT results and suggests that this agreement is not dependent on how the data is categorized (*Pusterla et al., 2002*).

Based on in-house studies, the manufacturer reported an overall 95.2% agreement between the reference method and the SNAP-BAT in dogs and cats (*Holbrook, Roth-Johnson & Sheldon, 2005*). Although these authors did not report sensitivity, specificity, or statistical analysis of agreement, these results are similar to the results reported in the present study. Furthermore, a recent study confirmed that there were no significant differences in serum bile acid concentrations, as measured by a standard chemistry analyzer, among laboratories (*Nanfelt et al., 2012*). If methodology is consistent, it is likely that SNAP-BAT results would be similarly consistent with chemistry results from multiple institutions. It was beyond the scope of this study to determine accuracy of the SNAP-BAT for the diagnosis of liver disease based on results of liver biopsies, since this information was not always known or recorded in each case.

A secondary purpose of this study was to determine precision of the SNAP-BAT. Precision, or repeatability, of a measurement system is the degree to which repeated measurements under unchanged conditions show the same results. The present study attempted to determine precision of the SNAP-BAT by evaluation of intra-assay variability. Precision of the enzymatic method (the method used for most in-house chemistry analyzers) for determination of total serum bile acids has been reported to be satisfactory (*Tobiasson & Kallberg, 1980*). In the study reported here, intra-assay variability of the SNAP-BAT was minimal, although the results could not be interpreted statistically because three of the five samples had a uniform distribution. As noted above, the samples with a uniform distribution all had values >30 µmol/L, whereas the two samples that exhibited intra-assay variability had values between 5 and 30 µmol/L and therefore were fully quantified. This variability is expected because of the large range of values that can be classified as >30 µmol/L. Ideally, samples used for this portion of the study should have had more variance: for example, samples with values of less than 5 µmol/L should have been pooled into a single sample, and samples with values greater than 30 µmol/L should have been pooled into a single sample. However, this was not possible since the SNAP-BAT and enzymatic chemistry values of the sample were not known prior to thawing the samples.

Sample dilution is recommended for precision analysis, and use of diluent for that purpose has been reported for SNAP T4 tests in dogs and cats and for SNAP IgG testing in foals (*Kemppainen & Birchfield, 2006*; *Pusterla et al., 2002*). Dilution is not recommended

by the manufacturer for SNAP-BAT, however. In the early stages of the project we attempted dilution of samples with enzymatic results >30 μmol/L using a variety of diluents, including saline, sterile water, plasma, and the manufacturer's diluent. Analysis of all of these diluted samples resulted in an error message from the SNAP-BAT Reader or in no change in the result as compared with the undiluted sample. Therefore, we do not recommend attempting to further quantify results greater than 30 μmol/L using the SNAP-BAT by means of dilution.

Although the SNAP-BAT is highly accurate as a semi-quantitative test, it has several limitations. The SNAP-BAT does not quantify any result less than 5 μmol/L or greater than 30 μmol/L. Quantification of bile acid concentrations greater than 30 μmol/L may be useful for differentiating the etiology of the liver dysfunction. Congenital portosystemic shunts (CPSS) and PVH-MVD are related conditions in which there is a malformation of the macroscopic or microscopic portal circulation, respectively (*Winkler et al., 2003*; *Berent & Tobias, 2009*). Because treatment and prognosis for the two conditions vary significantly, it is critical to differentiate them (*Allen et al., 1999*; *Christiansen et al., 2000*). Definitive differentiation between the two conditions can only be achieved by combining liver biopsy results with the results of nuclear scintigraphy or other advanced imaging; however, recent data suggests that bile acid results can be used to assist in distinguishing the conditions. In one study, pre- and post-prandial concentrations in dogs with PVH-MVD were found to be significantly lower ($41 \pm 57$ μmol/L and $92.7 \pm 37.3$ μmol/L, respectively) than in dogs with CPSS ($159 \pm 103$ μmol/L and $250 \pm 148$ μmol/L, respectively) (*Allen et al., 1999*). Of 21 dogs diagnosed with PVH-MVD at University of Tennessee (based on appropriate histologic changes and lack of shunting on nuclear scintigraphy), preprandial and/or postprandial SBA were increased compared to the reference range in 20/21 dogs; however, preprandial SBA were <75 μmol/L in 89% of these dogs, and post-prandial SBA were <75 μmol/L in 81%. In 100 dogs diagnosed with CPSS from 2003 to 2009 at the University of Tennessee, 95% of dogs had preprandial or postprandial SBA >50 μmol/L; 93% had preprandial or postprandial SBA >75 μmol/L; and 76% had preprandial or postprandial SBA >100 μmol/L (RT Hodshon, 2011, unpublished data). A fully quantitative test would be more useful for differentiating between CPSS and PVH-MVD.

Other limitations of the SNAP-BAT include the inability to run more than 2 samples concurrently and the need to purchase the SNAP Reader to interpret the results. The SNAP Reader is required because, unlike the 3DX and 4DX SNAP tests, the colorimetric change for measurement of SNAP-BAT cannot be visually interpreted. Benefits of the SNAP-BAT include ease of use and fast, immediate, precise and highly sensitive results.

The study itself had several limitations. The population that provided samples was skewed toward dogs with increased bile acid concentrations, since samples were obtained from dogs undergoing bile acid testing because of suspected liver disease. This resulted in a lack of normal distribution to our population. The effects of serum hemolysis and lipemia, which have been previously shown to falsely decrease and increase bile acid concentrations, respectively, as measured by enzymatic assay (*Jensen, 1991*; *Solter, Hoffmann & Hoffman, 2000*), were also not directly assessed in this study.

## CONCLUSION

The SNAP-BAT has a substantial to near perfect level of agreement to the enzymatic chemistry analyzer, is easy to use, yields immediate results, and has little intra-assay variability, particularly for results greater than 30 µmol/L. While highly sensitive, it may yield false positive results compared to the enzymatic chemistry analyzer. Additionally, it is not able to quantify values greater than 30 µmol/L, limiting its utility in determining severity of liver disease.

### Funding

This study was funded by the Companion Animal Grant, University of Tennessee. The funders had no role in study design, data collection and analysis, decision to publish, or preparation of the manuscript.

### Grant Disclosures

The following grant information was disclosed by the authors:
Companion Animal Grant, University of Tennessee.

### Competing Interests

The authors declare there are no competing interests.

### Author Contributions

- Rachel L. Seibert conceived and designed the experiments, performed the experiments, analyzed the data, wrote the paper, prepared figures and/or tables.
- Karen M. Tobias contributed reagents/materials/analysis tools, reviewed drafts of the paper.
- Ann Reed analyzed the data, statistician.
- Karl R. Snyder contributed reagents/materials/analysis tools, provided clinical pathology input and assistance.

### Animal Ethics

The following information was supplied relating to ethical approvals (i.e., approving body and any reference numbers):

University of Tennessee, Institutional Animal Care and Use Committee (IACUC) Protocol Number: 2053-1011.

### Supplemental Information

Supplemental information for this article can be found online at http://dx.doi.org/10.7717/peerj.539#supplemental-information.

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
