# Peer review of "Evaluation of a semiquantitative SNAP test for measurement of bile acids in dogs"

_PeerJ, doi:10.7717/peerj.539_

## Round 0.1 · original submission · Minor Revisions

Both reviewers feel this work has merit and point out a number of questions which should be addressed point by point in a detailed reply letter.

Reviewer 1 ·

Basic reporting

No Comments

Experimental design

No Comments

Validity of the findings

No Comments

Additional comments

More information on the chemical mechanism of the SNAP test should be provided in the introduction. Comparison between quantitative and semi-quantitative test for serum bile acids should be provided in the discussion.

Reviewer 2 ·

Basic reporting

The article was well-presented with standardized structure, with sufficient explanation of the goal of the study, experimental design, and interpretation of results.

Experimental design

The experiments were designed carefully with sufficient description in methods and data analysis. The only suggestion I have is that the authors can add a study using allotments of same sample and assay them in different days to test stability of the assays as a function of time.

Validity of the findings

The data is statistically sound and controlled, the conclusion was appropriately stated and connected to the original goal proposed.

---

## Round 0.2 · Major Revisions

Sorry for the delay as we were waiting a long time on one reviewer.

Two reviewers have now looked at this revision (the 2nd reviewer has been unable to supply comments due to computer issues, but I have been in correspondence with them).

Both reviewers feel that the article is well-written, with sufficient background, description of method, results description and appropriate discussion. The concern is that the present paper only compared two established, even commercialized methods for bile acid detection. If authors could categorize these 56 "dog patients" to show their bile acid values using the new method for normal and the abnormal bile acids levels associated with these "dog patients" at the different stages of the disease, this would be a good manuscript.

Reviewer 2 ·

Basic reporting

The article is written in an appropriate language and format, with sufficient background, description of method, and results. The only concern that the reviewer has is that the present manuscript only conducted the comparison of two established, even commercialized methods for bile acid detection. The novelty and significance of the present study is greatly limited.

Experimental design

The experiments were design properly.

Validity of the findings

The data is statistically sound and controlled.

---

## Round 0.3 · accepted · Accept

Thanks for carefully addressing reviewer's comments and revising the manuscript accordingly. We are looking forward to receiving your future work with this unique SNAP test.